# Adhesive Resins with High Shelf-Life Stability Based on Tetra Unsaturated Monomers with Tertiary Amines Moieties

**DOI:** 10.3390/polym13121944

**Published:** 2021-06-11

**Authors:** Alma Antonia Pérez-Mondragón, Carlos Enrique Cuevas-Suárez, Jesús García-Serrano, Nayely Trejo-Carbajal, A. Lobo-Guerrero, Ana M. Herrera-González

**Affiliations:** 1Doctorado en Ciencias de los Materiales, Universidad Autónoma del Estado de Hidalgo, Pachuca-Tulancingo Km. 4.5, Carboneras, Mineral de la Reforma 42184, Hgo., Mexico; alma.15.8.26@hotmail.com; 2Laboratorio de Polímeros, Instituto de Ciencias Básicas e Ingeniería, Universidad Autónoma del Estado de Hidalgo, Pachuca-Tulancingo Km. 4.5, Carboneras, Mineral de la Reforma 42184, Hgo., Mexico; jserrano@uaeh.edu.mx (J.G.-S.); nayeli_trejo@uaeh.edu.mx (N.T.-C.); azdlobo@gmail.com (A.L.-G.); 3Laboratorio de Biomateriales Dentales, Área Académica de Odontología, Instituto de Ciencias de la Salud, Universidad Autónoma del Estado de Hidalgo, Circuito Ex Hacienda La Concepción S/N, San Agustín Tlaxiaca 42160, Hgo., Mexico

**Keywords:** adhesives, resins, photopolymerization, mechanical properties, aging

## Abstract

This work reports the use of two monomers with two tertiary amines and four methacrylic (TTME) or acrylic (TTAC) terminal groups as co-initiators in the formulation of experimental resin adhesive systems. Both monomers were characterized by FT-IR and ^1^H NMR spectroscopies. The control adhesive was formulated with BisGMA, TEGDMA, HEMA, and the binary system CQ-EDAB as a photo-initiator system. For the experimental adhesives, the EDAB was completely replaced for the TTME or the TTAC monomers. The adhesives formulated with TTME or TTAC monomers achieved double bond conversion values close to 75%. Regarding the polymerization rate, materials formulated with TTME or TTAC achieved lower values than the material formulated with EDAB, giving them high shelf-life stability. The degree of conversion after shelf simulation was only reduced for the EDAB material. Ultimate tensile strength, translucency parameter, and micro-tensile bond strength to dentin were similar for control and experimental adhesive resins. Due to their characteristics, TTME and TTAC monomers are potentially useful in the formulation of photopolymerizable resins for dental use with high shelf-life stability.

## 1. Introduction

Adhesive systems in dentistry are used to achieve the adhesion of restorative materials to dental structures [1]. The use of restorative materials together with the adhesive technique has become common in dental practice, especially because professionals prefer these materials due to their advantages such as aesthetics, improved adhesive properties, and preservation of the dental structure, which, in turn, leads to the strengthening of the remaining dental structure [2]. 

Dental adhesive systems generally contain dimethacrylate monomers, organic solvents, and a photo-initiator system. The most used photo-initiator system contains camphorquinone (CQ) as a photo-initiator and ethyl-4-dimethylaminobenzoate (EDAB), a tertiary amine, as a co-initiator [3]. CQ produces free radicals when exposed to blue light with a wavelength of 450–500 nm, however, due to its chemical structure, they have short time of life. Therefore, the CQ requires a co-initiator, with tertiary amines being the most used compounds, which act as hydrogen donors, facilitating the generation of free radicals to provide the reactive radicals that begin polymerization [4]. However, the co-initiator that does not react during the polymerization process may be directly related to the biological properties of the polymer [5]. In this regard, it has been shown that penetration into the dentinal tubules of unpolymerized components can trigger inflammatory processes in the pulp of the tooth [6]. To overcome these drawbacks, new alternatives for the composition of dental adhesives have been studied. Some amines reported as co-initiators have terminal double bonds, which allow their binding to the resin and prevent their migration. The reported co-initiators have one to three terminal double bonds of the acrylic [3], methacrylic [4], or allylic [5] type, successfully replacing the EDAB co-initiator. Furthermore, with these co-initiators, double bond conversion [6] and flexural strength [4] have been improved. 

Considering this, the objective of this study was the evaluation of the TTME and TTAC monomers in the formulation of experimental adhesive resins without EDAB. The use of TTME and TTAC results in similar properties to the control adhesive resin formulated with the conventional co-initiator (EDAB). Additionally, the experimental monomers have two amino groups and four terminal double bonds within their structure. The amino groups allow them to act as co-initiators, while the four double bonds guarantee that all of the initiator forms part of the adhesive resin, preventing its migration. In addition, the TTMEE and TTAC co-initiators cause a decrease in the polymerization rate, and therefore the experimental resins have high shelf-life stability.

## 2. Materials and Methods

### 2.1. Materials and Instruments

The dimethacrylate monomers bisphenol A-glycidyl methacrylate (BisGMA), triethylene glycol dimethacrylate (TEGDMA), and (hydroxyethyl)methacrylate (HEMA), camphorquinone (CQ), and ethyl 4-(dimethylamino)benzoate (EDAB) used in the formulation of adhesive resins and 4,4′-methylenebis (N, N-diglycidylaniline), methacrylic acid, and acrylic acid used in the synthesis of the monomers were purchased from Sigma-Aldrich (St. Louis, MI, USA). The FT-IR spectra were obtained with a Perkin Elmer Frontier spectrophotometer using the reflectance (ATR) technique (Perkin Elmer, Waltham, MA, USA). The solvents used were distilled with the techniques described in the literature. The NMR spectra were obtained with a 400 MHz spectrometer (Bruker 1400, Inc. Palo Alto, CA, USA) using deuterated chloroform as a solvent and tetramethylsilane as an internal reference. The photopolymerization of the materials was carried out using a Bluephase N photopolymerization unit (Ivoclar-Vivadent) with an average irradiance of 1200 mW/cm^2^. The irradiance was periodically monitored using a digital radiometer (Bluephase Meter, Ivoclar-Vivadent). Mechanical properties were evaluated using a universal Instron 4465 mechanical testing machine (Instron, Norwood, MA, USA). Confocal microscopy images were obtained with a confocal laser scanning microscope (LSM 980 with Airyscan 2; Zeiss, Germany). For each methodology, the sample size was estimated based on the data of previous studies, considering a comparative study design with three independent groups, a power of 0.8, and α = 0.05.

### 2.2. Synthesis of Monomer TTMEE and TTAC

The TTMEE and TTAC were synthesized according to a methodology described elsewhere [7]: 1.0 g (2.36 mmol) of 4,4′-methylenebis (N,N-diglycidylaniline) was dissolved in 80 mL of ethyl acetate and placed in a two-necked round-bottom flask equipped with a magnetic stirring bar and a reflux condenser, under constant agitation. Next, 1.2 mL (28.4 mmol) of methacrylic acid, for TTME, or acrylic acid for TTAC, 2 wt.% of triethylamine and 5 ppm of hydroquinone were added (Figure 1). The reaction was carried out at reflux temperature for 24 h. For monomer purification, the ethyl acetate was evaporated, and 100 mL of dichloromethane was added, then three washes were carried out with a saturated solution of sodium carbonate, and after three washes with a 5% HCl solution. The organic phase was dried over anhydrous sodium sulfate, and then the solvent was removed and the monomers were obtained. Figure 1 shows the structure of the monomers. 

TTMEE characterization. FT-IR/ATR (cm^−1^): 2962 (νC-N), 1692 (νC=O), 1623 (νC=C), 1523 (νC=C_aromatic_). ^1^H-NMR (400 MHz, CDCl_3_): 7.05 (4H, m, H_B_), 6.72 (4H, m, H_C_), 6.20 (4H, s, H_G_), 6.20 (4H, s, H_G_), 5.6 (4H, s, H_H_), 3.70 (8H, m, H_F_), 3.40 (4H, m, H_E_), 3.15 (2H, m, H_A_), 2.75 (4H, m, H_D_), 2.55 (4H, m, H_D′_), 1.92 (12H, m, H_I_).

TTAC characterization. FT-IR/ATR (cm^−1^): 2922 (νC-N), 1732 (νC=O), 1623 (νC=C), 1504 (νC=C_aromatic_). ^1^H-NMR (400 MHz, CDCl_3_): 7.01 (4H, m, H_B_), 6.67 (4H, m, H_C_), 6.44 (4H, m, H_I_), 6.07 (4H, m, H_G_), 6.20 (4H, m, H_G_), 5.88 (4H, m, H_H_), 4.06 (4H, m, H_F_), 3.64 (4H, m, H_F′_), 3.34 (4H, m, H_E_), 3.12 (2H, m, H_A_), 2.76 (4H, m, H_D_), 2.62 (4H, m, H_D′_).

### 2.3. Formulation of Adhesive Resin

The control and experimental adhesive resins were formulated according to the composition presented in Table 1. The formulation of control adhesive resin was made by mixing the dimethacrylate monomers BisGMA, TEGDMA, and HEMA with the binary system CQ-EDAB as a photo-initiator. For the experimental groups, the EDAB was completely replaced for the TTME or the TTAC monomers, totaling two experimental and one control group. The concentration of TTME or the TTAC was determined using a screening test, evaluating the influence of its concentration on the degree of conversion of the experimental adhesive resin (n = 3). According to this screening test, the concentration of 25 wt.% was chosen.

### 2.4. Real-Time Polymerization Kinetics

A Perkin Elmer Frontier model infrared spectrophotometer equipped with an ATR cell was used to measure the degree of conversion and polymerization rate. For degree of conversion, a small sample (n = 3) of the unreacted mixture was placed in the cell and an infrared spectrum was obtained. Subsequently, the sample was irradiated for 30 s, and finally another infrared spectrum was obtained. In each of the spectra, the height of the absorption band of the aliphatic νC=C bond was measured at 1638 cm^−1^ and the height of the absorption band of the aromatic νC=C bond was located at 1609 cm^−1^. The degree of double bond conversion (*DC*) was determined according to the following equation [8]:(1)$$DC\left(\%\right)=\left(1-\frac{h_{1638}/h_{1609}\;pol}{h_{1638}/h_{1609}\;mon}\right)\times 100\%$$
where *h*_1638_ is the maximum height of the band at 1638 cm^−1^, and *h*_1609_ is the maximum height of the band at 1609 cm^−1^. The term “*mon*” corresponds to the spectrum of the unpolymerized monomer mixture, and the term “*pol*” refers to the spectrum of the polymerized material.

For polymerization rate measurements, the spectrophotometer software package was used in the monitoring scan mode in the range of 1500–1800 cm^−1^, a resolution of 4 cm^−1^, and a mirror speed of 2.8 mm/s. Within this configuration, one scan was acquired every 1 s during photoactivation. The degree of conversion per second was determined using the formula mentioned above. Then, the degree of double bond conversion (*DC*) vs. polymerization reaction time data was plotted, and Hill’s 1 three-parameter nonlinear regression was performed for curve fitting. The rate of polymerization, Rp, was calculated considering these data/fitted plots [9].

### 2.5. Degree of Double Bond Conversion After Shelf-Life Simulation 

To assess materials’ stability, the degree of conversion was evaluated initially and after 6 (T1), 18 (T2), and 24 (T3) months of shelf-life simulation by storing the materials in an incubator at 37 °C for different periods of time [10]. The time period necessary to achieve such periods was calculated using the Arrhenius model:(2)$$r={Q_{10}}^{\left(RT-ET/10\right)}$$
where *r* is the accelerated aging rate, *RT* is the storing temperature recommended by the manufacturers (4 °C), *ET* is the storage temperature in the acclimatization chamber (37 °C), and *Q*_10_ is the reaction rate coefficient (2).

For each period, the degree of conversion was determined following the methodology described above.

### 2.6. Sensitivity to Ambient Light

Sensitivity to ambient light [11] was determined by monitoring the degree of conversion of the materials after exposure to simulated ambient light. Ambient light was simulated using a dimmable light source, which was adapted to a Fourier transform infrared spectrophotometer equipped with an attenuated total reflection device (Frontier, Perkin Elmer, Waltham, MA, USA). The amount of light reaching the diamond crystal of the spectrophotometer was set at 8000 lx, measured using a digital luxmeter. To perform the test, a small sample of the unpolymerized material was placed in the ATR diamond cell and an infrared spectrum was obtained. Then, the sample was exposed for 60 s with the dimmable light source, and after that, an infrared spectrum was obtained again. The percentage of double bond conversion was calculated with the formula previously described. The experiment was performed three times for each group.

### 2.7. Ultimate Tensile Strength (UTS)

Bulb-shaped specimens with dimensions of 10 mm long and 1 mm thick were made using a metal mold. The samples were polymerized for 20 s on both sides using an LED-type photopolymerization unit. Once polymerized, the samples were stored in distilled water in a dark environment at 37 °C for 24 h. After that, the specimens were fixed to a metal jig using a cyanoacrylate-based adhesive and their tensile strength was evaluated in a universal mechanical testing machine (Instron 4465, Norwood, MA, USA) at a crosshead speed of 0.5 mm/min until the fracture. The cross-sectional area of each specimen was measured before testing using a digital caliper. The UTS was calculated, in MPa, by dividing the tensile force at failure (N) by the cross-sectional area of the specimen (mm^2^) [12]. 

### 2.8. Color Alteration and Translucency Parameter

Sixty disc-shaped samples (5 mm in diameter and 2 mm in thickness) were made from resin composite (Filtek^TM^ Z250, 3M ESPE, Sao Paulo, MN, USA) with the aim of simulating a direct restoration. A single increment of resin composite was placed in a silicon mold, pressed between 2 glass plates and light-cured for 20 s from the upper and the bottom surface using an LED photopolymerization unit, Bluephase N (Ivoclar-Vivadent, Schaan, Liechtenstein). The specimens were stored in a dark environment at 37 °C for 24 h to ensure stabilization of the polymer network. Afterwards, composite discs were randomly reassigned to 3 subgroups according to the adhesive system. Composite discs were placed into a 2.2 mm thick silicone matrix and the adhesive systems were applied in a single increment into the matrix. This configuration allowed to produce a 0.2 mm thick adhesive layer. Immediately after the application of the adhesive system, a Mylar^®^ strip covered by a glass slide was placed and the material was photoactivated [13]. 

Composite-adhesive disks were then removed from the silicon mold and the CIELAB color parameters were measured using a spectrophotometer (SP60, X-Rite, Grand Rapids, Miami, FL, USA). All specimens were measured over white (*L** = 93.1, *a** = 1.3, *b** = 5.3) and black (*L** = 27.9, *a** = 0, *b** = 0) standard tiles provided by the manufacturer. Color variation (Δ*E*_00_) was calculated according to the CIEDE2000 method using the following equation [14]:(3)$$\Delta E_{00}={\left[{\left(\frac{\Delta L^{\prime }}{K_{L}S_{L}}\right)}^{2}+{\left(\frac{\Delta C^{\prime }}{K_{C}S_{C}}\right)}^{2}+{\left(\frac{\Delta H^{\prime }}{K_{H}S_{H}}\right)}^{2}+R_{T}\left(\frac{\Delta C^{\prime }}{K_{C}S_{C}}\right)\left(\frac{\Delta H^{\prime }}{K_{H}S_{H}}\right)\right]}^{\frac{1}{2}}$$
where Δ*L*′, Δ*C*′, and Δ*H*′ are the differences of luminosity, color, and intensity in each specimen compared to the control in CIEDE2000, and *R_T_* is a rotational function that quantifies the color and intensity differences in the blue regions. The functions *S_L_*, *S_C_*, and *S_H_* adjust the total color variation in relation to the group treated in *L*′, a′, b′, and the parametric factors *K_L_*, *K_C_*, and *K_H_* are corrections by the experimental conditions. The translucency parameter (TP) for each specimen was calculated using the formula: TP = [(L*w − L*b)^2^ + (a*w − a*b)^2^ + (b*w − b*b)^2^]^1/2^, where w and b refer to the color coordinates measured on the white and black backgrounds.

### 2.9. Micro-Tensile Bond Strength (µ-TBS)

Extracted human third molars were used after approval of the Ethical Review Board from the School of Medical Sciences at the Autonomous University of Hidalgo State (protocol CEEI-032-2019). Immediately after extraction, all blood and adhered tissue were removed, and thereafter stored in 0.5% chloramine/water solution at 4 °C for one week. Then, teeth were stored in distilled water at 4 °C for no longer than two months until their use [15]. 

For specimen preparation, the root was sectioned, and the crowns were embedded in acrylic resin, allowing the buccal enamel surface to be exposed. Then, the occlusal part of the crowns was removed with an orthodontic grinder until exposure of a flat mid-coronal dentin surface. The exposed dentin surface was then wet-abraded using 600-grit silicon carbide sandpaper for 30 s to produce standardized smear-layers.

Dentin specimens were randomly divided into three groups based on the adhesive system used (n = 5). Before applying the adhesive resins, a commercial self-etch primer (Clearfil SE Bond 2 primer, Kuraray Noritake Dental Inc., Tokyo, Japan) was applied according to the manufacturer’s instructions. Then, the corresponding adhesive resins were applied and rubbed for 10 s into the surface and photoactivated for 10 s. After the bonding procedures, resin composite build-ups (Filtek^TM^ Z250) were constructed in 3 increments of 2 mm each, and each layer was polymerized for 30 s. Light-curing procedures were performed using an LED photopolymerization unit, Bluephase N (Ivoclar-Vivadent). After immersion in distilled water at 37 °C for 24 h, the specimens were sectioned using a slow-speed diamond saw (Isomet Saw 1000 Precision, Buehler Ltd., Lake Bluff, IL, USA) to obtain resin-dentin sticks with a cross-sectional area of approximately 0.9 mm^2^. After storage in distilled water at 37 °C for 24 h, the sticks were individually fixed to a tensile testing device with cyanoacrylate glue and the TBS was tested in a mechanical universal test machine (Instron 4465), at a crosshead speed of 1 mm/min with a 1000 N load cell.

The fractured portions of the specimens were observed under a light stereoscope at 40× magnification to classify failures as adhesive, cohesive within dentin, cohesive within composite, or mixed failures. For each tooth, five sticks were tested, and the results obtained were averaged and used for statistical purposes. Specimens with pretesting failures were included in the tooth mean value; for this, the average value between zero and the lowest bond strength value obtained in each tooth was used [15].

### 2.10. Confocal Microscopy

A qualitative analysis of the penetration depth of the adhesive resin into the dentine substrate and the hybrid layer formed was performed using confocal light scanning microscopy. Following the same adhesive protocols described above, one tooth per group was prepared for observation. Before application on the dentine surface, the fluorochrome Rhodamine B (Sigma Aldrich, St. Louis, MI, USA) was added into the respective adhesives at a concentration of 0.08%. After the bonding procedures, one composite increment was placed over the adhesive resin. Then, the teeth were cut longitudinally into two halves and the generated surfaces were polished for one minute with SiC paper in sequence (Grit 800/1200/2000/4000) under water cooling. The samples were examined in 1000-fold magnification under a CLSM (LSM 980 with Airyscan 2; Zeiss, Gina, Germany) at a 514 nm excitation line of the argon ion laser. The emissions were detected using a DD 458/514 band-pass filter.

### 2.11. Statistical Analysis

Statistical analysis was performed using Sigma Plot 12.0 software. The data were analyzed to verify the normal distribution and homogeneity of the variance. A one-way analysis of variance (ANOVA) was used to compare the dependent variables. In all cases, the level of significance was set at *p* = 0.05.

## 3. Results

TTAC and TTME were synthetized via a single-step synthetic route according to the literature. Their structure was confirmed by means of FT-IR and ^1^H-NMR spectroscopies. FT-IR spectra of these compounds revealed that the absorption band at 910 cm^−1^, corresponding to the epoxy group of the initial compounds, disappears, suggesting the opening of the epoxy ring. The latter is confirmed by the appearance of the elongation vibration of the hydroxyl group at 3428 cm^−1^. In addition, new absorption bands located at 1711 cm^−1^, assigned to the elongation vibration mode of the carbonyl group νC=O, and at 1629 cm^−1^, assigned to the elongation vibration of the νC=C bond of the methacrylic groups, suggest the incorporation of the acryloyl and methacryloyl functional groups within the structure of the raw material (Figure 2). 

With regards to ^1^H-NMR spectroscopy, the main evidence of the reaction between epoxy groups, the acid group of acryloyl or methacryloyl acid, can be found in the signals corresponding to the protons of the akene. For TTAC, the signals corresponding to this functional group appear as a multiple signal at 6.45 ppm, integrating four protons in the cis position of the double bond, and at 5.9 ppm, integrating four protons in the trans position of the double bond. On the other hand, for the TTME monomer, protons from the double bond appear as a singlet located at 6.20 ppm, integrating four protons, and as a singlet at 5.6 ppm, integrating four protons, in trans and cis position with respect to the methyl of the methacrylic group, respectively (Figure 3).

Polymerization kinetics for experimental and control groups were evaluated. According to the results from this test, both experimental materials exhibited a significant reduction in the R_pmax_ values (Figure 4A). 

The degree of double bond conversion after shelf-life simulation is shown in Figure 5. The degree of conversion of experimental adhesives formulated with TTAC or TTME monomers showed stability after the different periods of simulated shelf-life: 6 (T1), 18 (T2), and 24 (T3) months of shelf-life simulation, by storing the adhesive resins in an incubator at 37 °C [10]. Conversely, adhesive resin formulated with EDAB significantly reduced its degree of double bond conversion after the second (18 months) shelf-life simulation, indicating the low stability. 

The sensitivity to ambient light of each adhesive is shown in Table 2. According to the results, the material formulated with EDAB reached significantly higher values of degree of conversion after exposure to simulated ambient light. The ultimate tensile strength (UTS) of the experimental and control materials is also shown in Table 2. The results showed that differences in the means of the UTS among the materials were not statistically significant (*p* = 0.968).

Figure 6 shows the color alteration and the translucency parameter of the materials evaluated. The results suggest that the application of the adhesive systems resulted in a shift in DE from the original color of the composite to above the human perceptibility threshold [16]. Despite this, the difference in the DE shift was not statistically significant between the TTME and the control (*p* > 0.05), and in the case of the TTAC adhesive, the DE shift observed was statistically significantly lower (*p* < 0.05). Additionally, none of the adhesive systems tested in this study promoted statistically significant differences in the translucency of the composite (*p* = 0.893).

Table 3 shows the values of the micro-tensile bond strength (µ-TBS) to dentin and the failure mode of the adhesive resins tested. According to the results, no statistically significant differences were observed between the mean values of the bond strength among the groups (*p* = 0.820). 

Confocal light scanning microscopy images of the bonded interfaces are shown in Figure 7. No noticeable differences were detected in homogeneity and continuity of the adhesive layer along the interfaces observed. However, the experimental materials formulated with TTAC and TTME presented longer resinous tags than the EDAB adhesive.

## 4. Discussion

In this work, the synthesis of two tetramethacrylates, TTAC and TTME, was reported. They were used as co-initiators in the formulation of experimental adhesive resins. The monomers have two amino groups and four terminal double bonds within their structure. The amino groups allow them to act as co-initiators, while the four double bonds guarantee that all of the initiator forms part of the adhesive resin, preventing its migration. Their chemical–mechanical performance was compared with an adhesive resin formulated with EDAB, used as a control. Overall, TTAC and MBTTME achieved lower R_pmax_ values and similar degrees of conversion values compared to EDAB. After shelf-life simulation and sensitivity to ambient light, experimental adhesives showed better stability than the EDAB adhesive resin. With regards to UTS and bonding properties, they were similar to those presented by the EDAB adhesive. Considering this, the hypothesis was partially accepted.

According to Figure 4, both experimental materials exhibited a significant reduction in the R_pmax_ values (Figure 4A). This behavior suggests that free radical formation in experimental systems is not as fast as that which occurs in the system containing EDAB. Another possible explanation lies in the fact that, due to the TTAC and TTME binding to the polymer, vitrification of the material occurs at early stages, preventing any further extensive reaction [17]. This can be confirmed when plotting R_pmax_ against DC (Figure 4B), where it can be seen that, when compared to the control, R_pmax_ for the experimental materials is achieved at lower degree of conversion values. 

Despite the lower R_pmax_ values obtained, it is important to note that the experimental materials were able to reach double bond conversion values similar to the those of the control group, and in the case of the TTAC monomer, it was able to reach statistically significant higher values. This result helps to demonstrate that the increase of double bonds in the structure of the monomers could increase the crosslinking during polymerization reaction. Besides this, comparing both experimental adhesives, the adhesive that has acrylate groups achieved higher Rp values than the adhesive with methacrylate groups, because methacrylate groups have hyperconjugation, which causes less reactivity [18]. 

The degree of double bond conversion of experimental adhesives formulated with TTAC or TTME monomers showed stability after the different periods of simulated shelf-life. Conversely, the control material formulated with EDAB significantly reduced its degree of conversion after the second and third periods of the shelf-life simulation. The shelf-life of materials used in dental clinical practice is extremely important. The reason for restoration failures might not only be due to poor clinical procedures but also to the limited shelf-life of some of the materials used [10]. This study evaluated the shelf-life of the materials formulated through the determination of the degree of double bond conversion of the materials after different periods of simulated shelf-life. The results obtained suggested that the degree of conversion was reduced after shelf-life simulation only for the control material, whereas the materials formulated with TTAC and TTME showed excellent stability, where the latter could be due to the lower polymerization rate observed in these materials [19]. This behavior means that experimental adhesive resins could better retain the physical and mechanical properties necessary to accomplish its prescribed purpose [20].

The sensitivity to ambient light of each adhesive was tested to elucidate their ability to prevent the spontaneous polymerization of the material (Table 2). To evaluate this, no polymerization inhibitor was used in the formulation of the new adhesive materials [21]. According to the results, the material formulated with EDAB reached significantly higher values of degree of conversion after exposure to simulated ambient light, which means that handling of this material may be impaired by ambient and operatory light units during any process of the restorative procedure [22].

The mechanical properties of the adhesives were also evaluated. The mechanical property was measured in terms of ultimate tensile strength (UTS), a property that is frequently used to evaluate the performance of an adhesive in terms of its tensile bond strength with the local tissue [23]. The results showed that differences in the means of the UTS among the materials were not statistically significant (*p* = 0.968). The mechanical properties of polymers mostly depend on the adequate conversion of double bonds during polymerization [24]; besides this, the configuration and crosslinking of the polymer may have a great influence on the mechanical properties too [25]. In this sense, variations in the concentration and type of monomers in the formulation of the adhesive systems plays an important role, and this can explain why the material formulated with the TTAC monomer, even with higher values of degree of conversion, showed statistically similar UTS.

Some optical parameters of the experimental adhesives were also evaluated (Figure 6). The new adhesive systems were applied under a 2 mm thick composite simulating a direct restoration. According to the results, the application of the adhesive systems resulted in a shift in DE from the original color of the composite to above the human perceptibility threshold [16]. This is in agreement with other studies that have demonstrated that the adhesive system plays an important role in changing the color of direct composite restorations [26,27]. Despite this, it is important to note that the differences in the DE shift observed were not statistically significant between the TTME and the control materials (*p* > 0.05), and in the case of the TTAC adhesive, the DE shift observed was lower (*p* < 0.05). Additionally, none of the adhesive systems tested in this study promoted statistically significant differences in the translucency of the composite (*p* = 0.893), so it can be suggested that the incorporation of the TTAC or TTME monomer into an experimental adhesive resin would not be clinically perceivable.

Finally, the micro-tensile bond strength (µ-TBS) to dentin and the failure mode of adhesive resins were tested using in vitro test methods that provide reliable data for materials’ development and/or evaluation of experimental variables. The micro-tensile bond strength test is currently recommended as the best method to evaluate the bond strength of adhesive systems, and it is considered useful for preliminary evaluation as a pre-clinical test [15]. According to Table 3, no statistically significant differences in the mean values of the bond strength among the groups were observed. In this sense, under clinical conditions, the use of the TTAC or TTME monomers for the formulation of adhesive resins would result in optimal retentive strength and marginal seal and bonding efficacy, rendering a similar performance and durability as the control in the clinical service of restorations.

## 5. Conclusions

The TTME and TTAC monomers were successfully used as polymerizable co-initiators in the formulation of experimental resin adhesive systems. The characteristics shown by the experimental materials can prove that the use of TTME and TTAC as polymerizable co-initiators represents an adequate alternative for the development of new resin-based materials. TTME and TTAC could provide high shelf-life stability and extended working time compared to the conventional materials formulated with EDAB. 

## Figures and Tables

**Figure 1 polymers-13-01944-f001:**
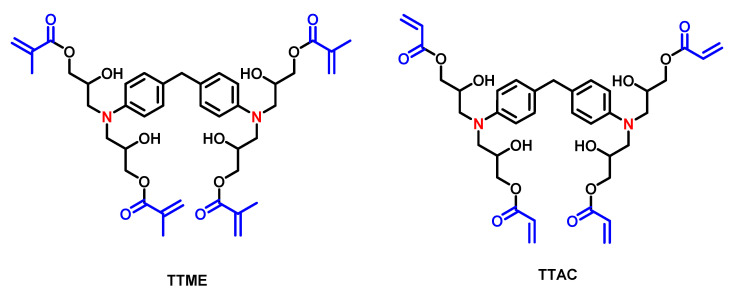
Chemical structure of tetra unsaturated monomers TTME and TTAC, evaluated as a co-initiator. Unsaturation (blue) and tertiary amines (red).

**Figure 2 polymers-13-01944-f002:**
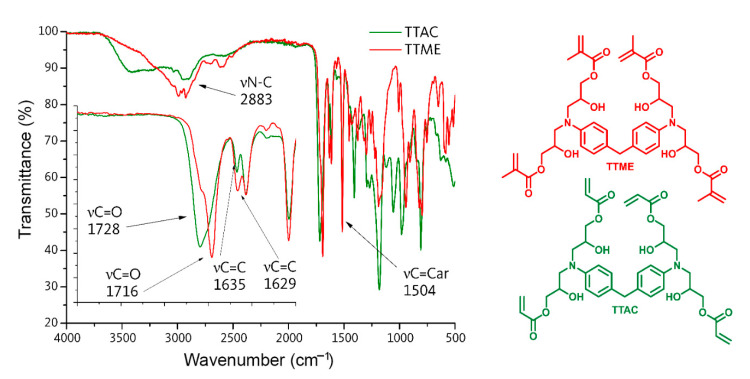
FT-IR spectra of monomers TTA and TTME.

**Figure 3 polymers-13-01944-f003:**
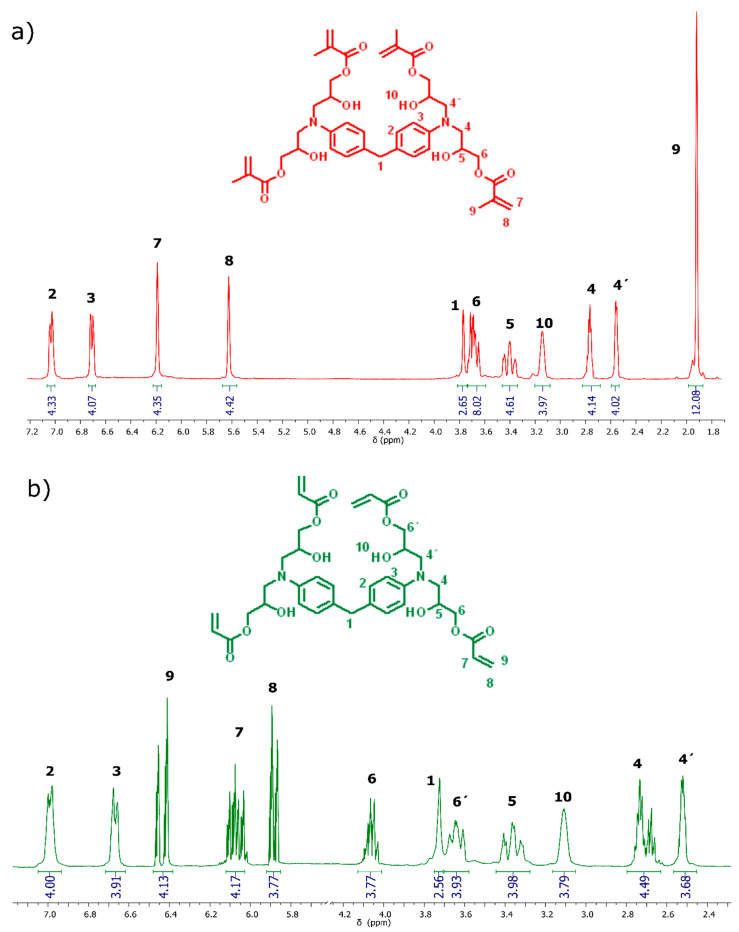
^1^H-NMR spectra of the monomers (**a**) TTME and (**b**) TTA.

**Figure 4 polymers-13-01944-f004:**
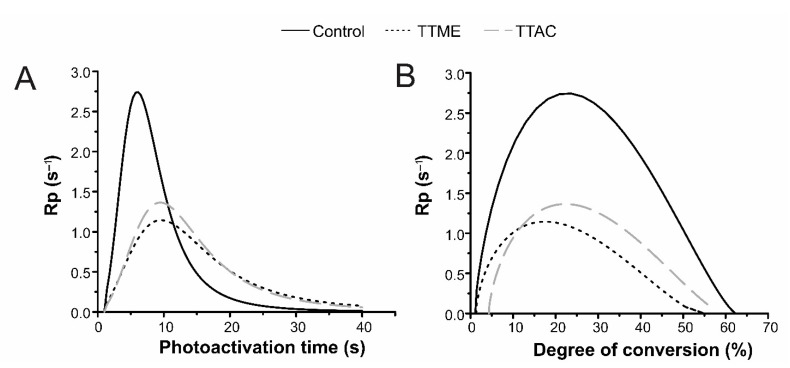
Polymerization kinetics of the formulated adhesive resins. (**A**) Rp as a function of time and (**B**) Rp as a function of degree of conversion.

**Figure 5 polymers-13-01944-f005:**
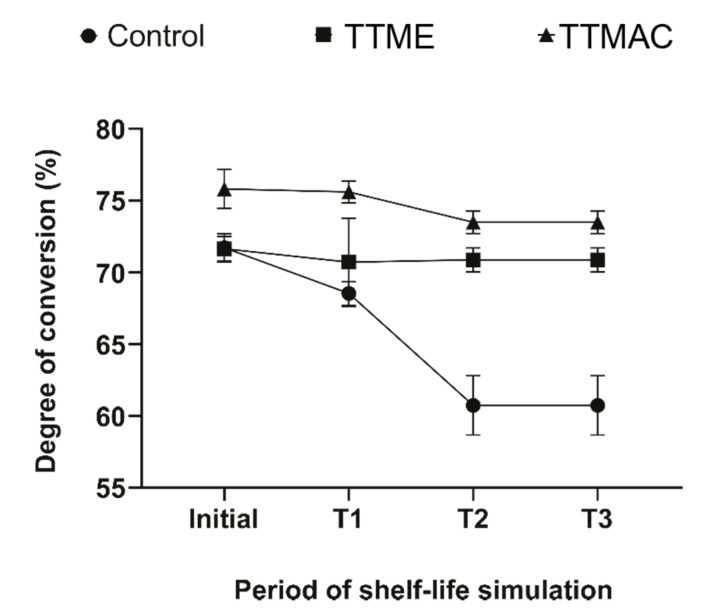
Degree of double bond conversion of adhesive resins after shelf-life simulation.

**Figure 6 polymers-13-01944-f006:**
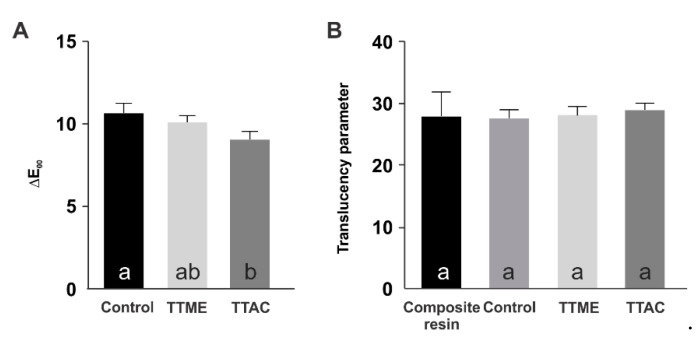
Color alteration (**A**) and translucency parameter (**B**) of the adhesive resins evaluated.

**Figure 7 polymers-13-01944-f007:**
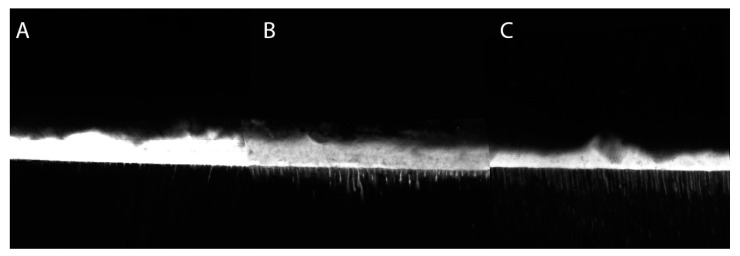
Confocal light scanning microscopy of the bonded interfaces of the tested groups. (**A**) EDAB adhesive, (**B**) TTME adhesive, and (**C**) TTAC adhesive.

**Table 1 polymers-13-01944-t001:** Composition of experimental adhesive resin.

Group	Formulation wt.%						
	BisGMA	TEGDMA	HEMA	TTME	TTAC	CQ	EDAB
EDAB	50	25	25	-	-	0.5	1
TTME	40	20	20	20	-	0.5	-
TTAC	40	20	20	-	20	0.5	-

**Table 2 polymers-13-01944-t002:** Mean (standard deviations) for degree of conversion, sensitivity to ambient light, ultimate tensile strength, and crosslinking percentage.

Adhesive Resin	Degree of Double Bond Conversion (%)	Sensitivity to Ambient Light (%)	Ultimate Tensile Strength (MPa)
EDAB Adhesive	71.71 (0.97) ^b^	7.57 (0.21) ^a^	9.42 (1.40) ^a^
TTME Adhesive	71.63 (0.86) ^b^	0.64 (0.24) ^b^	9.65 (2.78) ^a^
TTAC Adhesive	75.80 (1.35) ^a^	0.31 (0.24) ^b^	9.64 (2.06) ^a^

Values followed by distinct superscript letters indicate significant differences in the columns for experimental adhesive resins tested (*p* < 0.05).

**Table 3 polymers-13-01944-t003:** Means (standard deviations) for the micro-tensile bond strength (µ-TBS) to dentin and the failure mode of the adhesive resins tested.

Experimental Group	µ-TBS (MPa)	Failure Mode (%)				
		Cohesive within Composite	Cohesive within Dentin	Adhesive	Mixed	Pre-Testing Failure
EDAB Adhesive	16.33 (2.04) ^a^	9	9	26	52	4
TTME Adhesive	17.23 (1.98) ^a^	4	20	32	32	12
TTAC Adhesive	17.14 (3.19) ^a^	4	25	21	42	8

Values followed by distinct superscript letters indicate significant differences in the columns for the experimental adhesive resins tested (*p* < 0.05). Variability in the frequency of failure mode for the adhesives was statistically significant (*p* = 0.006).

## Data Availability

The data that support the findings of this study are available from the corresponding author upon reasonable request.

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
