# Peer review of "Adhesive Resins with High Shelf-Life Stability Based on Tetra Unsaturated Monomers with Tertiary Amines Moieties"

_polymers, 2021, doi:10.3390/polym13121944_

Round 1

Reviewer 1 Report

The paper is devoted for adhesive resins with high shelf-life preparation and investigations. The topic is generally interesting, however the paper contains a lot of unexplained places and need major revisions.

Page 2 lines 64-65, please explain what is the relation between „the polymerization rate“ and „shelf-life  stability“?

Why namely CQ-EDAB wasused as photoinitiator?

Infrared and NMR investigations of your samples should be demonstrated in the paper and properly discussed with corresponding references, not only briefly mentioned.

Fig. 2B is not commented or cited in the paper text.

Page 9 line 342 „light scanning microscopy images of the SEM“ is is correct?

Results presented in Table 3 should be more discussed.

DSC investigations of your samples would be very useful.

Author Response

1.    Page 2 lines 64-65, please explain what is the relation between „the polymerization rate“ and „shelf-life  stability“?

R. This was briefly discussed in the Discussion section. Thank you!

Text Change:

Page 12. Line 404.

… The results obtained suggested that degree of conversion was reduced after shelf-life simulation only for the control material, whereas the materials formulated with TTAC and TTME showed excellent stability, the last could be due to the lower polymerization rate observed in these materials [19]… 

[19: Moszner, N.; Salz, U. Recent developments of new components for dental adhesives and composites. Macromol. Mater. Eng. 2007, 292, 245–271, doi:10.1002/mame.200600414.]

2.    Why namely CQ-EDAB was used as photoinitiator?

R. The camphorquinone-tertiary amine was used because is the most common photoinitiator system in resin-based materials (DOI: 10.3390/polym13030470).

3.    Infrared and NMR investigations of your samples should be demonstrated in the paper and properly discussed with corresponding references, not only briefly mentioned.

R. Thank you for the suggestion. Infrared and H1 NMR characterization of the samples was added and discussed. 

4.    Fig. 2B is not commented or cited in the paper text.

R. Figure 2B (now is 4B) is cited and commented at page 11 line 382.

5.    Page 9 line 342 „light scanning microscopy images of the SEM“ is correct?

R. You are right, sorry for the mistake. The term SEM was deleted. Thank you!

6.    Results presented in Table 3 should be more discussed.

R. Thank you for the suggestion. Results in Table 3 were more discussed now. 

Text change:

Page 11. Lines 440 and 447-449.

Finally, the micro-tensile bond strength (µ-TBS) to dentin and failure mode of adhesive resins were tested. In vitro test methods that provide reliable data for materials development and/or evaluation of experimental variables. The micro-tensile bond strength test is currently recommended as the best method to evaluate the bond strength of adhesive systems, and its considered useful for preliminary evaluation as a pre-clinical test [15]. According to Table 3, no statistically significant differences in the mean values of the bond strength among the groups were observed. In this sense, under clinical conditions, the use of the TTAC or TTME monomers for the formulation of adhesive resins would result in optimal retentive strength, marginal seal and bonding efficacy, rendering a similar performance and durability than the control in the clinical service of restorations.

7.    DSC investigations of your samples would be very useful.

R. We agree with you regarding that DSC characterization will bring us more information to better describe the mechanical and chemical performance of the newly synthesized material. Unfortunately, we do not have the equipment to carry out this characterization. However, the mechanical properties at room temperature indicated that the new adhesive have good mechanical properties. And for this application the materials need good performance at corporal temperature. 

Reviewer 2 Report

This paper reports on the “Adhesive resins with high shelf-life stability based on tetra unsaturated monomers with tertiary amines moieties”. The article is interesting. Introduction, methodology and reference, results and discussion seems be corrected.

I have few comments to the manuscript:

  1. Materials and instruments. Speared.
  2. FT-IR. Missing figure (in results) and interpretation (in instrumental).
  3. Paragraphs 2.4 – 2.9. Missing reference.

Taking into account all comments the manuscript may be published in Polymers after minor revision.

Author Response

I have few comments to the manuscript:

1.    Materials and instruments. Speared.

R. We are so sorry; we cannot understand this comment.

2.    FT-IR. Missing figure (in results) and interpretation (in instrumental).

R. Infrared and H1 NMR characterization of the samples was added and discussed.

3.    Paragraphs 2.4 – 2.9. Missing reference.

R. The following references were added into the manuscript. 

2.4 Real time polymerization kinetics (DOI: 10.1016/j.dental.2019.11.008)

2.5 Degree of double bond conversion after shelf-life simulation (DOI: 10.1016/j.dental.2019.05.023)

2.6 Sensitivity to ambient light (DOI: ISO 4049)

2.7 Ultimate tensile strength (DOI: 10.1186/s40563-019-0120-0)

2.8 Color alteration and translucency parameter (DOI: 10.5301/jabfm.5000277)

2.9 Micro-tensile bond strength (µ-TBS) (DOI: 10.1016/j.dental.2016.11.015)

4.    Taking into account all comments the manuscript may be published in Polymers after minor revision.

R. Thank you for your time for reviewing this manuscript.

Round 2

Reviewer 1 Report

Authors make proper corrections according to referee remarks.

I suggest to publish the paper as it.

This manuscript is a resubmission of an earlier submission. The following is a list of the peer review reports and author responses from that submission.

Round 1

Reviewer 1 Report

The authors present an interesting study in conception. However, the manuscript is too rare. The manuscript has too many methodological and formal errors to be accepted or considered. Some critical points that the authors should take into consideration:
-The authors must review its title, it is too daring with the few experiments carried out. The abstract presented by the authors is rare and does not allow the reader to have an idea that makes it attractive.
-The authors should think about restructuring their introduction. The introduction presented by the authors is too rare and does not focus on the possible novelty of the molecules they intend to implement.
-The material and methods is absolutely unjustifiable. Authors should adequately describe the methodology in terms of formulation and fluorometry. Figure 1 should include the structure indicating the modified parts. In this sense, authors should describe all figures adequately with more complete figure legends.
-The formula 3 must be reviewed, it has conceptual errors.
-The sample size is not adequate. This point must be justified, it is too small.
-The statistical analysis is not adequate. ANOVA is too permissive in such a small sample analysis. The authors must think that this point is very limiting in their study.
-The results are described in a superficial way. Figure 3 shows T1 / T2 / T3 ??? justify the different T. This point is not adequately justified in the discussion.
-Figure 4 should be reviewed, because it does not allow the results to be justified. Authors must include the exact error bars. Please, authors should include results as IQR. This point will allow the reader to have an exact idea of ​​your deviations and their results.
-Figure 5 should be adequately described. Because the sample size is so limited, authors must include all the images they have. This point can allow authors to improve their results.
-The discussion is too pretentious. Authors should discuss the state of the art, and propose a real and translational proof of concept that they intend to implement.
-The authors must review their conclusions, where they must include and justify all the limitations that their study has.
-Authors should extensively review their manuscript in English. You would invite the authors to use the editing services.

Reviewer 2 Report

In the manuscript “Adhesive resin composition with high shelf-life stability based on tetramethacrylate monomers” submitted to “Polymers” for potential publication, authors have investigated two monomers with tertiary amines and four methacrylic or acrylic terminal unsaturations as co-initiators in the formulation of experimental resin adhesive systems and characterized using FT-IR and 1H NMR spectroscopies.

The study requires some major improvements in the design and presentation:

Abstract require re-writing as the objective is not clear and presented a lot of abbreviations. Key results are not presented in the abstract.

Introduction is very short and background is completely missing to support the objective and rationale of the study

In the composition table, all constituents in the rows for each group don't make exact 100 wt% showing imbalance of the ingredients in each group for the base resins like bisgma tegdma and hema in each group.

Mainly, the discrepancies in these resins will not give us a clear explanation. The study design should be corrected, and rationale of the study should be provided.